# The Optimal Axis-Symmetrical Plasma Potential Distribution for Plasma Mass Separation

**DOI:** 10.3390/molecules27206824

**Published:** 2022-10-12

**Authors:** Andrey Pavlovich Oiler, Gennadii Dmitrievich Liziakin, Andrey Vladimirovich Gavrikov, Valentin Panteleimonovich Smirnov

**Affiliations:** 1Joint Institute for High Temperatures, Russian Academy of Sciences, Moscow 125412, Russia; 2Moscow Institute of Physics and Technology, Dolgoprudny 141700, Russia

**Keywords:** active particle system, plasma potential distribution, mass separation, particle trajectories

## Abstract

The mass separation of chemical element mixtures is a relevant task for numerous applications in the nuclear power industry. One of the promising approaches to solve this problem is plasma mass separation. In a recent study, the efficiency of plasma mass separation in a configuration with a potential well and a homogeneous magnetic field was experimentally demonstrated. This article examines the possibility of increasing the distance between the deposition regions of charged particles with different masses by varying the profile of the electric field potential. Such correlation can be considered as the control in a system of active particles. A cylindrical coordinate system is used. The electric field is axially symmetrical, and the magnetic field is directed along the axis of the symmetry. The corresponding mathematical problem was solved in a general way. The criteria for increasing the distance between the deposition areas of the “light” and “heavy” components of the mixture have been formulated. A high sensitivity of particle trajectories to the electric field potential in the region of the pericentres of the trajectories of charged particles was detected. Recommendations for the practical implementation of the optimal spatial separation of ion fluxes are proposed.

## 1. Introduction

At present, there is considerable interest in active particle systems, which are thermodynamically open systems. In such systems, under certain conditions of interaction with the external environment, a significant change takes place in the behaviour of particles, as compared to equilibrium closed systems.

Recently, a whole research area devoted to plasma systems of active particles has emerged. These are both plasma systems with micron-sized macroscopic particles immersed in them [1,2,3,4] and more traditional plasma systems without a dust component [5,6]. Such plasma formations are open particle systems in which, due to specially selected external conditions, it is possible to transform chaotic thermal motion into directed flows of active particles.

If the system consists of charged particles of different masses, by using their activity, a mass spectrometer can be created based on the effective separation of components of different masses, for example, in low-temperature plasma [5,7,8].

This study is devoted to the investigation of another open plasma system of active particles, in which it is possible to form conditions that make it possible to effectively separate its elements by the magnitude of the charge-to-mass ratio. A condensed mixture of two substances is heated and thereby converted into vapour to create a plasma system. Thereafter, the vapour is ionised by an electron beam to a plasma state, which mainly includes single-charged ions.

One option to enable the components of the mixture to be separated is to pull ions separately from the source using electromagnetic fields (electromagnetic separation method) [9]. However, the particle flow obtained in this way is limited, due to the formation of a positive volumetric charge. Moreover, Coulomb repulsion greatly increases the spread of ion trajectories, which in turn complicates their active control.

Another method is to operate a plasma flow when there is a volume charge compensation. In fact, when the restriction is removed on the possible number of particles in the flow, the system behaviour acquires the characteristic features of active particle systems and the performance of the separation device, which uses the features of their movement, increases dramatically [10]. The resulting system of active charged particles (ions), which are in a background plasma, acquires kinetic energy due to interactions with electromagnetic fields created in the plasma volume. Varying the configuration of the electromagnetic field makes it possible to control the movement by self-adjusting the trajectories of particles due to their activity in an open system. This study is devoted to the investigation of the conditions under which the behaviour of these active charged particles can lead to the separation effect of the elements.

The main difficulty of controlling charged particles in plasma is to create a given plasma potential profile. When a charged body is placed in plasma, its potential is shielded at a distance of the Debye radius. This is due to the plasma polarisation. In this case, it will not be possible to create the necessary distribution of plasma potential. However, the presence of a magnetic field fundamentally changes the situation. The magnetic field restricts the transport of charged particles across the field lines. At the same time, along the magnetic field lines, the conductivity can be maintained at a high level. Then, the potential of the electrodes can propagate along the lines, and the potential drop will occur in the transverse direction [11]. The question as to the extent to which such a transverse field can be controlled remains open [12,13,14]. However, there are studies that indicate a significant effect of thermal electron emissions from end electrodes on the potential distribution in the plasma column [14,15,16,17,18].

Having created a specified distribution of the electrostatic potential, the flows of the injected mixture, consisting of active charged particles, can be spatially separated, and the result of such separation has been demonstrated experimentally [19]. This approach to separating elements into groups may be very useful to reprocess spent nuclear fuel (SNF).

The release of SNF into the environment, in the case of unsuccessful disposal, requires the emergency neutralisation of radioactive substances. For example, the sorption of actinides from aqueous solutions [20], since such contamination is a serious threat to human health. Reducing the amount of buried waste is one of the most important motivations for the implementation of a closed nuclear fuel cycle. The possibility of SNF processing and the separation of a mixture of substances, in turn, is one of the key tasks of such a closure. Currently used hydrometallurgical methods of reprocessing SNF involve the formation of concomitant liquid waste, the amount of which is difficult to reduce. In view of this problem, the “dry” methods of SNF processing such as fluoride volatility [21], pyroelectrochemical [22,23,24], and plasma methods [25,26,27] have received active development. The latter are modern developments of the aforementioned electromagnetic methods of mass separation, the principles of which were developed in the middle of the 20th century [9,28,29]. For ions with the same electric charge, but different masses, the heavy component will have more inertia than the light component, so the radius of the curvature of the trajectory movement of heavy particles will be greater than that of light particles. By separating the fission products from the uranium that has not yet decayed, it is possible to process SNF by separating the part for reuse (closure of the nuclear fuel cycle) and the part for burial.

Various configurations of plasma mass separators have now been achieved. Among them, one of the most developed separators is the Ohkawa mass filter [30], which is based on the incapability of rotating plasma to confine ions with mass that is higher than the critical one. This concept was performed in experimental studies [31] and in studies [32,33]. The other types are plasma–optical mass separators [34,35], in which separated particle beams with an initial rotation are injected into the radial electric field. The ratio between centrifugal and electrical forces determines the central mass. Additionally, mass separation in a configuration with a potential well is quite promising. The configuration of electromagnetic fields is chosen so that heavy particles fall into the potential well and light ones do not [19,36].

Plasma mass separation in the configuration of electromagnetic fields, with a potential well for ions in a homogeneous axial magnetic field, is investigated in Reference [19]. Figure 1 shows the main elements of such a separation scheme. With this approach, the entire volume of the installation is filled with background plasma. A negative offset is applied to the sectioned end electrodes. The potential of these electrodes propagates along the magnetic field lines and thus creates a radial electric field in the plasma. The injection of the separated particles occurs at the periphery of the plasma column. Positive ions (in this case, Ag and Pb ions) are attracted in a direction transverse to the magnetic field into a potential well formed in the plasma volume. The spatial separation of ions by mass occurs in the process of transportation. The separated substance is deposited on the substrate, which is located in accordance with the equipotential surface passing through the injection point. Within the framework of this approach, there are an infinite number of configurations of electric potential (potential well shapes), some of which are considered in References [36,37]. Ignoring the distortions introduced into the plasma volume by the structures of the substrate and the plasma injector of the separated substances, it can be assumed that the system has axial symmetry. Therefore, in the current study, consideration is limited to only axially symmetric configurations of the electric potential.

The trajectories of positively charged particles in crossed radial electric and axial magnetic fields have an epitrochoidal shape. First, the particles approach the axis of the system, and then, when they have gained speed, the magnetic field turns them in the direction of the periphery. Thus, the kinetic energy obtained due to the interaction with the background plasma is returned to plasma, equivalent to the energy lost through collisions. The substrate, collecting the separated substance, is positioned at the periphery of the discharge, where the trajectories of the particle beams make reversals. The radial velocity of the particles is minimal at the turning points, and as a result, the particles will settle better on the substrate.

When ions are separated in a reflective discharge with a thermionic cathode, the deposition regions of different elements partially overlap, as was shown in a recent experimental study [38]. The overlap occurs because of an increase in the dispersion (cross-section) of the beams in the components of the mixture. The possible factors leading to this are the scattering of particles by initial velocities and energies, distortions of the potential profile as a result of the injection, as well as fluctuations in the plasma potential [19,39]. Such a source of significant potential fluctuations in the Penning discharge could be a rotating structure, the characteristics of which have been described in Reference [40]. Based on these facts, it became necessary to increase the distance between the deposition areas of the components, in the mixture, to exclude the overlap of the beams in the limited space of the vacuum chamber. One of the most promising ways to improve mass separation is to find the optimal electric potential profile.

Numerical calculations of particle trajectories were carried out in linear and piecewise linear (consisting of two linear sections) axially symmetric potentials in the study [39]. It was shown that, with a constant radial electric field and the injection of particles at a distance of 25 cm from the axis, the characteristic separation scale was 10 cm. The separation scale can be significantly increased for a piecewise given potential profile; however, such a configuration is more demanding for the initial conditions.

The results clearly show that the shape of the potential could significantly affect the separation of ion fluxes with the same charges, but different masses. This prompted the question as to the existence and search for the “most effective” potential. In the current study, a more general problem is considered, which is devoted to finding conditions for the movement of ion fluxes of a separable mixture of active particles, leading to more efficient separation. The search for the optimal potential distribution is carried out among the entire class of axially symmetric electric field configurations. The only limitation that is imposed on the shape of the potential is its monotonicity, i.e., the absence of negative derivatives of the electric potential function.

To simplify the model, the energy losses due to collisions with the background discharge plasma will be considered negligible in this study, and thus it is assumed that the active interaction of the beam particles with the background plasma occurs only by an electromagnetic field.

## 2. The Determination of the Angular Coordinate of the First U-Turn Point

Consider the equation of motion of an ion with a charge q=Ze (Z is the charge number, e is the magnitude of the elementary charge) and mass $m=Am_{p}$ (A is the atomic number, $m_{p}$ is the mass of a proton) in a magnetic field $\vec{B}$ directed along the axis of a cylindrical vacuum chamber and an electric field $\vec{E}$ directed to the axis everywhere (Figure 2). The designations of the coordinate axes coincide with the axes in Figure 1. Let the injection point have a radius vector ${\vec{r}}_{0}$ and let the particle start from this position with zero initial velocity:(1)$$\{\begin{array}{c} m\ddot{\vec{r}}=q\left(\vec{E}+\dot{\vec{r}}\times \vec{B}\right);\\ \vec{r}\left(0\right)={\vec{r}}_{0};\;\dot{\vec{r}}\left(0\right)=0. \end{array}$$

Here, $\vec{r}$ is the radius vector to the current position of the ion. The electric potential at the injection point is assumed to be zero $\varphi \left(r_{0}\right)=0$.

Introduce the following dimensionless variables: potential $\Phi =\varphi /\varphi_{0}=\varphi {\left(eB^{2}r_{0}^{2}/m_{p}\right)}^{-1}$, radial coordinate $\rho =r/r_{0}$, and time $\tau =\Omega_{c}t$, where $\Omega_{c}=eB/Am_{p}$ is the cyclotron frequency. Then, the dimensionless Equation (1) in polar coordinates, the plane of which is perpendicular to the magnetic field, is written as follows:(2)$$\{\begin{array}{c} \ddot{\rho }-\rho {\dot{\theta }}^{2}=-\frac{A}{Z}\frac{d\Phi }{d\rho }+\rho \dot{\theta };\\ \rho \ddot{\theta }+2\dot{\rho }\dot{\theta }=-\dot{\rho }. \end{array}$$
(3)$$\{\begin{array}{c} \theta \left(0\right)=0;\dot{\theta }\left(0\right)=0;\\ \rho \left(0\right)=1;\dot{\rho }\left(0\right)=0;\\ \Phi \left(1\right)=0. \end{array}$$

Here, the dots denote derivatives with respect to dimensionless time. Next, the ions are considered to be single-charged Z=1. The integration of Equations (2), taking into account the initial conditions (3), gives an expression for the angular and radial velocities of ions:(4)$$\{\begin{array}{c} \dot{\theta }=\frac{1}{2}\left(\frac{1}{\rho^{2}}-1\right);\\ \dot{\rho }=\sqrt{-2A\Phi \left(\rho \right)-\rho^{2}{\dot{\theta }}^{2}}. \end{array}$$

Denote $\rho_{i}$ (i={1,2}—notation for light and heavy components. 1—light, 2—heavy) as the radial coordinate at which the radial velocity of an ion with atomic mass $A_{i}$ turns to zero for the first time after the start of the motion. The points of particle trajectories with radial coordinates $\rho_{i}$ are the closest to the centre of the coordinate system (see Figure 3). For brevity, they will hereafter be called the pericentres of the corresponding trajectories. The angular coordinate of the first U-turn point on the periphery can be obtained by integrating the derivative of the angular coordinate along the radial one, taking into account the symmetry of the problem:(5)$$\theta_{i}=2\int_{\rho_{i}}^{1}\frac{d\theta }{d\rho }d\rho =2\int_{\rho_{i}}^{1}\frac{\dot{\theta }}{\dot{\rho }}d\rho ;$$

In Equation (5), there is a multiplier of two, since the particle first gains an angular coordinate when it moves to the centre, and then when it moves from the centre. Substituting Equation (4) into (5), one can get:(6)$$\theta_{i}=2\int_{\rho_{i}}^{1}\frac{\dot{\theta }}{\dot{\rho }}d\rho =\int_{\rho_{i}}^{1}\frac{2\left(1-\rho^{2}\right)d\rho }{\sqrt{-8A_{i}\Phi \left(\rho \right)\rho^{4}-\rho^{2}{\left(1-\rho^{2}\right)}^{2}}}=\int_{\rho_{i}}^{1}f\left(\Phi \left(\rho \right),A_{i},\rho \right)d\rho .$$

Note that, at $\rho =\rho_{i}$, the root expression in (6) turns to zero. For further analysis, the next replacement will be introduced as:(7)$$\Phi \left(\rho \right)=-\frac{1}{8}{\left(\frac{1}{\rho^{2}}-1\right)}^{2}P\left(\rho \right),$$

Then Equation (6) will take the form:(8)$$\theta_{i}=\int_{\rho_{i}}^{1}\frac{2d\rho }{\sqrt{A_{i}P\left(\rho \right)-\rho^{2}}}.$$

It should be noted that the function P(ρ) must be positive on any segment of integration. Otherwise, a positively charged particle will not be able to overcome the potential barrier. According to Equation (8), the radial coordinate of the pericentre can be found using the equation $A_{i}P\left(\rho_{i}\right)=\rho_{i}^{2}$. The potential at which the root expression is identically equal to zero is:(9)$$B_{i}\left(\rho \right)={\Phi \left(\rho \right)|}_{P\left(\rho \right)=\rho^{2}/A_{i}}=-\frac{1}{8}{\left(\frac{1}{\rho^{2}}-1\right)}^{2}\frac{\rho^{2}}{A_{i}}.$$

This potential will be called the Brillouin potential for the i-th kind of particles [41]. Brillouin introduced this potential as the electric potential of stationary rotating electron beam without radial electron movement. This rotation differs from well-known rigid body rotation with the Brillouin limit because these cases have different boundary conditions. As Brillouin’s paper [41] states, electrons have zero velocity on the cathode edge, and his boundary conditions are similar to Equation (3), and consequently, our problem is considered analogically, but we deal with positive charged particles.

## 3. The Analysis of the Angular Distance between Beams

To study the separation process, the trajectories of particles with different atomic masses Ai will be analysed. The separation quality is determined by the value of the angular distance Δθ between the points of the first U-turn of the particles (Figure 3). Assume $A_{2}>A_{1}$. Then, due to greater inertia, everything else is equal, the trajectory of the heavy component will bend less under the influence of external fields, and its trajectory will approach closer to the coordinate centre (axis of the vacuum chamber) compared to the trajectory of a light particle, in other words, $\rho_{2}<\rho_{1}$. A rigorous proof of this fact is given in Appendix A. The expression for the angular distance can be written as follows by rearranging the integral expressions in such a way as to obtain integrals with disjoint integration segments.



(10)
$$\begin{array}{c} \Delta \theta =\theta_{2}-\theta_{1}=\int_{\rho_{2}}^{1}\frac{2d\rho }{\sqrt{A_{2}P\left(\rho \right)-\rho^{2}}}-\int_{\rho_{1}}^{1}\frac{2d\rho }{\sqrt{A_{1}P\left(\rho \right)-\rho^{2}}}\\ =\int_{\rho_{2}}^{\rho_{1}}\frac{2d\rho }{\sqrt{A_{2}P\left(\rho \right)-\rho^{2}}}-\int_{\rho_{1}}^{1}\left(-\frac{2d\rho }{\sqrt{A_{2}P\left(\rho \right)-\rho^{2}}}+\frac{2d\rho }{\sqrt{A_{1}P\left(\rho \right)-\rho^{2}}}\right) \end{array}$$



It is easy to notice that the condition $A_{2}>A_{1}$ implies strict positivity of the integrand in the second integral in the last form of Equation (10).

There are two ways to increase Δθ: Either to increase the value of the first integral or to decrease the value of the second, which in this case is equivalent to the corresponding operations with the values of integral functions for any point on the corresponding segment of integration. The “closer” (Figure 4) the function P(ρ) is to the parabola $\rho^{2}/A_{2}$ on the interval $\left(\rho_{2},\rho_{1}\right)$ (the smaller the functional norm of the difference of functions), the larger the first integral is. The “further” (greater the norm of difference) the function P(ρ) is from the parabola $\rho^{2}/A_{1}$ on the interval $\left(\rho_{1},1\right)$, the smaller the second integral, and consequently the greater Δθ is. Thus, it turns out that by selecting the function P(ρ), and hence the potential φ(r), it is possible to obtain any predetermined value of the angular distance between the points of the first U-turn of the beams of separated particles. Geometrically, the U-turn points are located on a circle that is equipotential with the potential of the injection point. Then, the maximum removal of groups of separated particles from each other is achieved at an angle of 180°, i.e., Δθ=π, since this is the maximum angle between the directions to the points located on the circle.

Using the example of Δθ=π, consider the function P(ρ) in the form of a polynomial of the second degree $P\left(\rho \right)=a\rho^{2}+b\rho +c$ on which the conditions $A_{1}P\left(\rho_{1}\right)=\rho_{1}^{2}$, $A_{2}P\left(\rho_{2}\right)=\rho_{2}^{2}$, are imposed, as well as the condition on the derivative at the point $\rho =\rho_{2}$: $A_{2}P^{\prime }\left(\rho_{2}\right)=2\rho_{2}+\chi$. Here, χ is a free parameter, selecting which one can get the necessary value Δθ=π for any set of given $\rho_{1}$ and $\rho_{2}$. The value Δθ=π for this class of functions is obviously achievable, because when χ tends to zero, the first integral in (8) will tend to infinity, since when χ=0 it diverges. In total, an uncountable set of radial profiles of the plasma potential is obtained, at which the maximum separation of an ion mixture consisting of two components is achieved.

Currently, lead (heavy component) and silver (light component) are used as model substances for the development of plasma separation technology in separation experiments [19]. The value of the axial magnetic field is B=1400 G. An example of a function for these substances in a given magnetic field at selected values is $\rho_{1}=0.8,\;\rho_{2}=0.3$ (the required parameter value χ≈0.087) and the resulting trajectories are shown in Figure 4.

## 4. The Formulation and Solution of the Variational Problem

It is necessary to choose a potential for practical implementation that, according to expectations, can be recreated in the experiment since there are infinitely many potential profiles for maximum beam separation. In this regard, it seems reasonable to start the search with a potential that minimally differs from the potential experimentally realised earlier [19]. Therefore, consider the following variational problem with respect to a dimensionless potential:(11)$$\mathrm{get}\;\Phi \left(\rho \right):F\left[\Phi \right]={\left(\theta_{2}\left[\Phi \right]-\theta_{1}\left[\Phi \right]-\pi \right)}^{2}\to min.$$

Here, $\theta_{i}\left[\Phi \right]$ is the functional defined by Equation (6). To formulate the idea differently, there is an uncountable set of absolute minima of functional (11). Therefore, the task is reduced to finding the absolute minimum in the vicinity of a given point. As mentioned earlier, another constraint placed on the desired potential profiles is the need for monotonicity in the profile, i.e., the absence of negative derivatives of the potential along the radius.

The task is solved numerically. The segment [0,1] is divided into N parts of length h, and the radial profile of the potential is replaced by the *N*+1 value at the nodes of the partition. Thus, the functional in (11) is replaced by a function of N variables—potential values (Figure 5):(12)$$F\left(\Phi \right)\to F\left(\Phi_{0},\Phi_{1},\ldots ,\Phi_{N-1}\right)=F\left(\vec{\Phi }\right).$$

According to the boundary condition $\Phi_{N}=0$, $\Phi_{N}$ is not a variable.

The search for the $F\left(\vec{\Phi }\right)$ minimum was performed using the steepest descent method [42] (for more details, see Appendix B). In Reference [19], the radial potential profile is quasi-linear with an electric field from 10 to 20 V/cm. Therefore, a linear potential with an electric field of 15 V/cm was taken as an initial approximation for calculations. The difference in the angular distance between the particle collection points for the initial potential profile was Δθ=0.470 rad, the iterative process was stopped at Δ*θ* = 3.095 rad.

The calculation results are shown in Figure 6. As can be seen from the figure, the result of minimising the functional (11) differs from the initial profile in two narrowly localised places at r=7.2 cm (r=7.1973 cm is radial coordinate of Ag pericentre) and at r=9.6 cm (r=9.6202 cm is radial coordinate of Pb pericentre) with a characteristic width of 10 µm and an amplitude of changes in the order of 5 mV. Such a modification in the potential profile is not available for practical implementation because the scale is too small. However, the above results indicate the critical importance of the areas with the specified coordinates. Any plasma fluctuations in this region significantly affect the movement of particles, and consequently, the result of the separation of ions with different masses and the same charge.

## 5. The Sensitivity to the Potential in the Pericentres of Trajectories

The analysis of the results shown in Figure 6 showed that two localised places, the modification of which could significantly improve the quality of separation, corresponded to the radial coordinates of the pericentres of the heavy and light components. Note that, in recent experiments, Pb was used as a heavy component and Ag was used as a light component [19]. In this regard, further reasoning is carried out for these elements. To assess the sensitivity of the angular separation to the potential, a local change in the potential by the magnitude $\delta \Phi_{k}$ in the vicinity of the point $\rho_{k}$ with width h is considered. Then, based on Equation (6), by the mean theorem, the change in the angular coordinate of the U-turn point can be estimated:(13)$$\delta \theta_{i}\approx h\delta \Phi_{k}{\frac{\partial f\left(\Phi ,A_{i},\rho_{i}\right)}{\partial \Phi }|}_{\rho =\rho_{k}}.$$

According to Equation (6), the asymptotics of the derivative of the angular coordinate of the U-turn point in potential will be:(14)$$\frac{\partial \theta_{i}}{\partial \Phi_{k}}\underset{\rho \to \rho_{k}}{\to }\frac{C_{i}}{{\left(\rho -\rho_{i}\right)}^{3/2}},\;C_{i}=const.$$

This derivative has power-law asymptotics when the radial coordinate tends to the radial coordinate of the pericentre. The closer to the pericentre, the greater the changes that will be caused by the variation of the plasma potential.

The corresponding modifications of the plasma potential profile should be in the order of tenths of the maximum values of the problem for practical implementation, i.e., the order of tens of volts and several centimetres. As already shown, the separation result is very sensitive to changes in the potential in the pericentres of the separated substances, so it is logical to vary the potential in this area. To simulate local potential changes, consider a family of curves representing the sum of a linear profile and a Gaussian function with a characteristic width σ and a maximum at the point $\rho =\rho_{0}$:(15)$$\Phi_{modified}\left(\rho \right)=\frac{15}{3377}\left(\rho -1+a\frac{\sigma }{\sqrt{2}}exp\left(\frac{1}{2}-\frac{{\left(\rho -\rho_{0}\right)}^{2}}{\sigma^{2}}\right)-a\frac{\sigma }{\sqrt{2}}exp\left(\frac{1}{2}-\frac{{\left(1-\rho_{0}\right)}^{2}}{\sigma^{2}}\right)\right).$$

The amplitude parameterisation is chosen so that the parameter a takes values from −1 to 1. For all the values a∈(−1,1), the derivative of the potential along the radius does not change the sign, and a point with a zero field appears at the extreme values a=±1 (Figure 7).

A numerical study was carried out for these two types of ions to solve the practical problem of separating a mixture of lead and silver. Using arguments similar to those described in Section 3, it can be stated that the further the plasma potential is from the Brillouin potential for silver on the segment from the starting point to the silver pericentre (the intersection of the potential with the Brillouin potential), the greater the angular distance between Ag and Pb will be. Additionally, the closer the plasma potential is to the Brillouin potential for lead, at the radial coordinate in the interval between the pericentres of the components of the separated mixture, the greater this angular distance will be. Thus, the maximum angular distance is achieved with the parameter a=1 (Figure 7).

Figure 8 shows the dependence of the angular separation of beams on the characteristic width and position of the local change in the potential (15) with the parameter *a* = 1. It can be seen from the figure that by varying the values of $\rho_{0}$ and *σ*, it is possible to obtain the separation of particles with any predetermined angle up to a value of 5 radians. However, for practical implementation, one should choose such $\rho_{0}$ and *σ*, the variation of which in a certain range of values will not lead to a significant deterioration in the separation. In other words, if one selects $\rho_{0}$ and *σ* corresponding to the red area in Figure 8, then due to the high sensitivity Δθ to changes in the values of $\rho_{0}$ and *σ* in this area, the errors accompanying the experimental implementation of such injection can lead, in practice, to a significant decrease in the separation angle.

From a practical point of view, the optimal choice may be a potential profile with a characteristic value of the parameters $\rho_{0}$ and *σ* shifted to a region of lower sensitivity Δθ for these parameters, for example, a profile with a “plateau” width at the level of $r_{0}\sigma =2.7\;\mathrm{cm}$ and the position of its centre in the coordinate of $r_{0}\sigma =10.8\;\mathrm{cm}$ and a potential in the order of −50 V. Figure 9 shows the trajectories of lead and silver ions in a potential type (15) with these parameters.

Figure 9 illustrates that movement in a weak electric field in the vicinity of the pericentre of the trajectory of light ions leads to their earlier reflection. Heavy ions overcome the potential plateau by inertia, and then accelerate in the region of an increased electric field. It can also be noticed that the radial coordinate of the lead pericentre has not changed much, but for silver it has noticeably increased.

## 6. Discussion

This paper analysed the trajectories of charged particles in crossed electric and magnetic fields when injecting a particle from the periphery of the studied system, which has cylindrical symmetry. This rather general mathematical problem was considered in relation to plasma mass separation. An expression was obtained for the angular coordinate of the first rotation of the particle at the periphery, depending on the spatial distribution of the axially symmetric electric potential of the plasma. It was shown that, for components of a particle flow with differing atomic masses, any predetermined angular separation (including Δθ=π) could be achieved, and this could be done in an infinite number of ways. The variational problem of calculation of the local minimum of the functional in the vicinity of the initially given potential and the strong sensitivity of the angular separation of components of the particle beam of different masses from the electric potential values in the vicinity of the pericentres of their trajectories, was found. The study concluded that realistic variations should be in the vicinity of pericentres. Based on these results, a recommendation was formulated for the experimental implementation of plasma mass separation. A previous study, proposed to create a local potential plateau, in which the light component of the mixture cannot overcome, but the heavy component can. Such a plateau is proposed to be set using end electrodes. Due to the sensitivity of particle trajectories for potential profile changes near pericentres, it is recommended to make this plateau wide enough. It helps to reduce harmful effects from errors while setting potential profile parameters. This is demonstrated in Figure 8, wherein small changes of parameters do not cause dramatic effects. In addition, the plasma potential profile can be distorted by plasma potential fluctuations, which can also affect particle trajectories. Angular separation changes due to potential fluctuations depend on shape, amplitude and the period of deviation. This makes the analysis more complicated, and therefore requires further research.

## Figures and Tables

**Figure 1 molecules-27-06824-f001:**
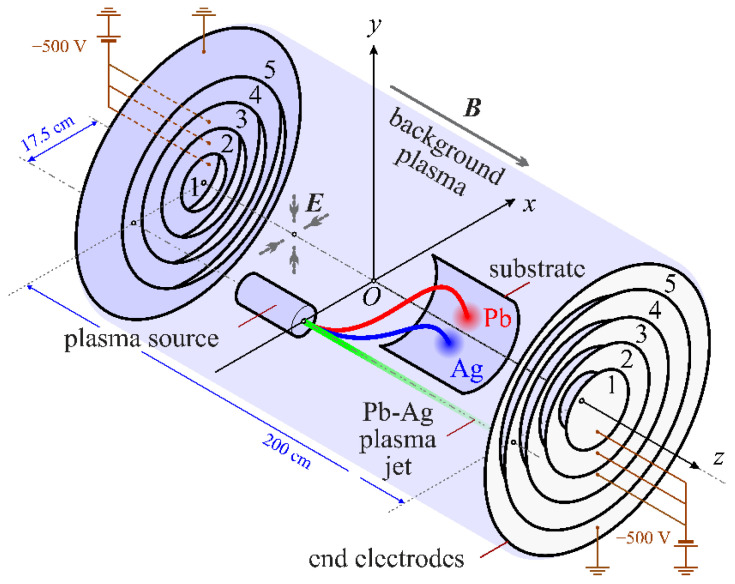
Schematic diagram of plasma mass separation in a configuration with a potential well [19]. Reprinted with permission from Ref. [38]. 2021, © IOP Publishing Ltd., Bristol, UK.

**Figure 2 molecules-27-06824-f002:**
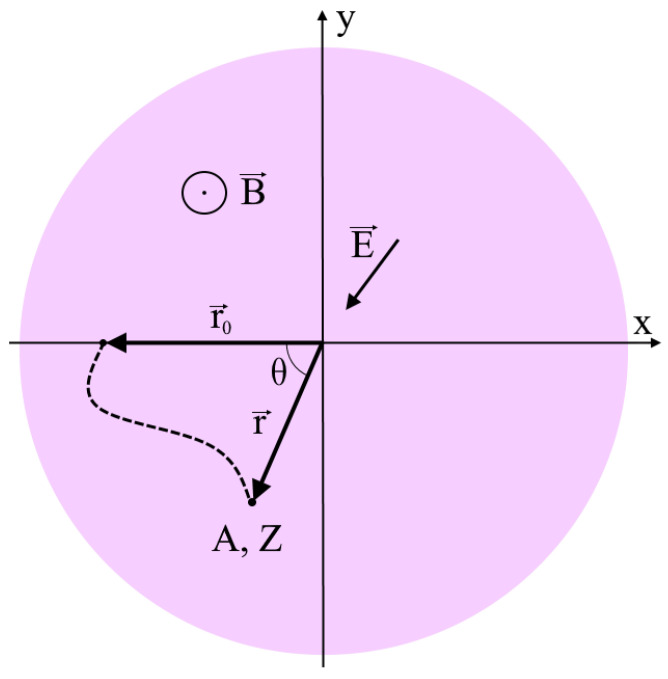
The basic designations of a mathematical problem. A and Z are the atomic and charge numbers of the ion, respectively. In the process of movement, the position of the ion is characterized by an azimuthal angle θ and a polar distance r.

**Figure 3 molecules-27-06824-f003:**
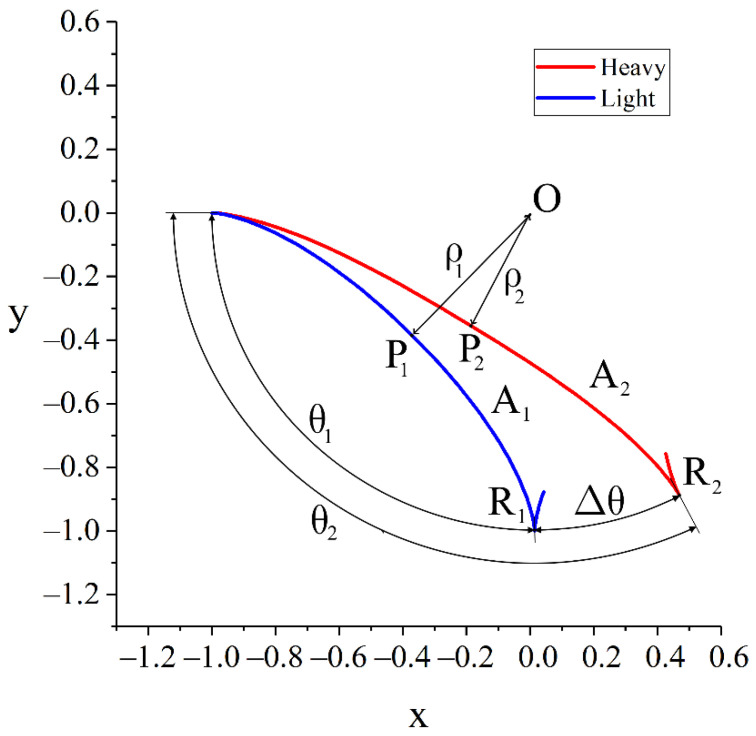
Determination of the parameters of particle trajectories. $A_{i}$ is the atomic number; $\theta_{i}$ is the angular coordinate of the first U-turn point $R_{i}$; and $\rho_{i}$ is the radial coordinate of the trajectory point (pericentre) $P_{i}$ closest to the axis of symmetry O. Index 1 refers to the light component, index 2 refers to the heavy component.

**Figure 4 molecules-27-06824-f004:**
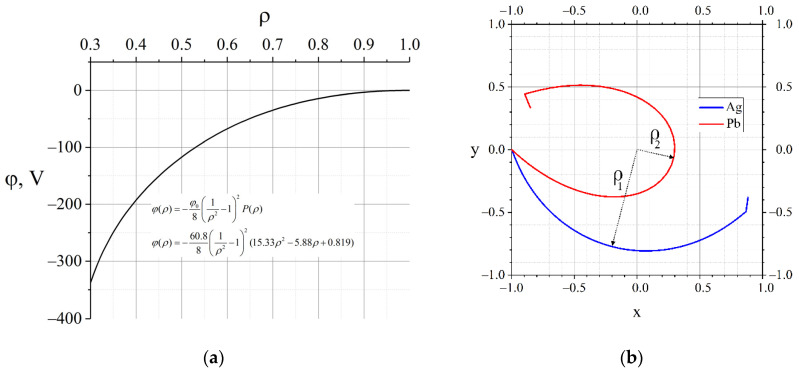
(**a**) An example of the radial profile of the potential where Δθ=π is achieved; (**b**) the trajectories of lead and silver in the specified potential.

**Figure 5 molecules-27-06824-f005:**
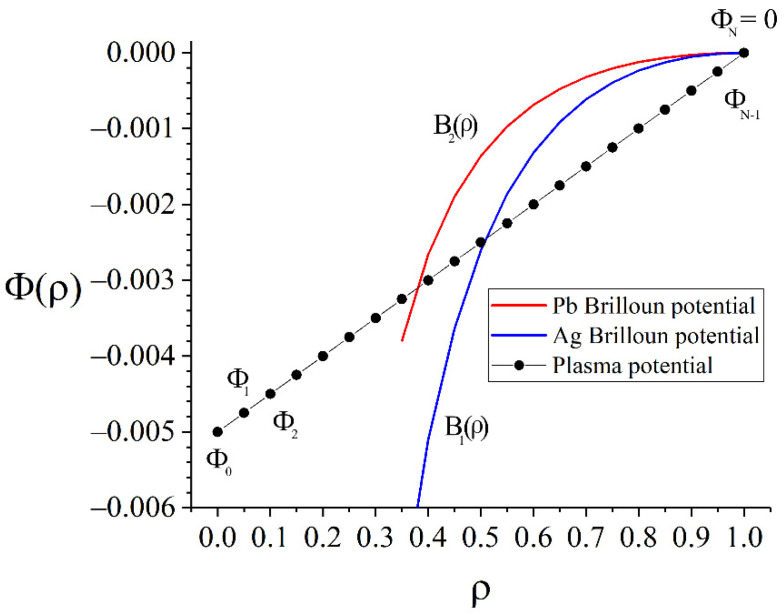
The definition of $F\left(\vec{\Phi }\right)$ as a function of many variables. Potential $\vec{\Phi }$ values at different radial coordinates act as independent variables. The intersection of a given potential with the Brillouin potential (Equation (9)) for a given variety of particles sets the radial coordinate of the trajectory pericentre.

**Figure 6 molecules-27-06824-f006:**
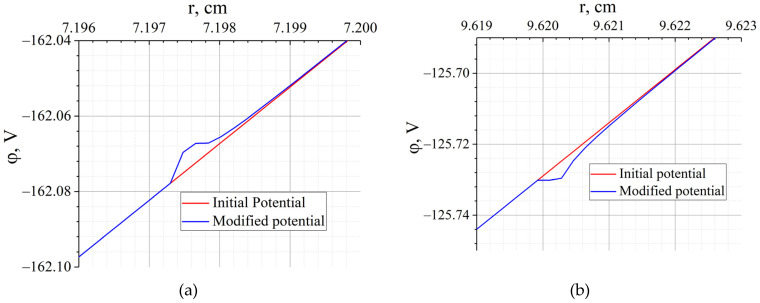
The result of $F\left(\vec{\Phi }\right)$ minimizing (blue line) when using as an initial approximation a linear potential with a gradient of 15 V/cm (red line): (**a**) in the vicinity of *r* = 7.2 cm and (**b**) in the vicinity of *r* = 9.6 cm.

**Figure 7 molecules-27-06824-f007:**
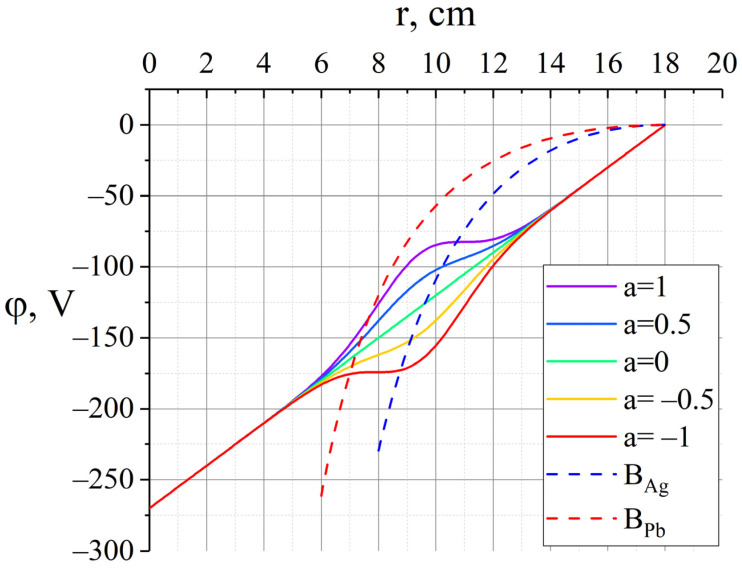
A family of curves simulating macroscopic changes in the plasma potential with a width σ=0.12 in the vicinity of the point *ρ*_0_ = 0.95 cm and their comparison with the Brillouin potentials (9) for Pb and Ag.

**Figure 8 molecules-27-06824-f008:**
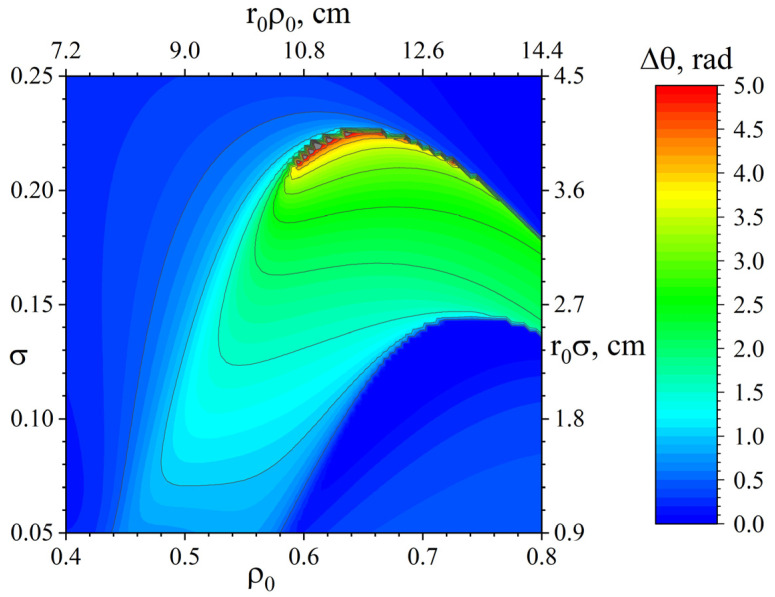
Dependence of the angular separation of lead and silver on σ and *ρ*_0_ for the family of curves (15) at *a* = 1.

**Figure 9 molecules-27-06824-f009:**
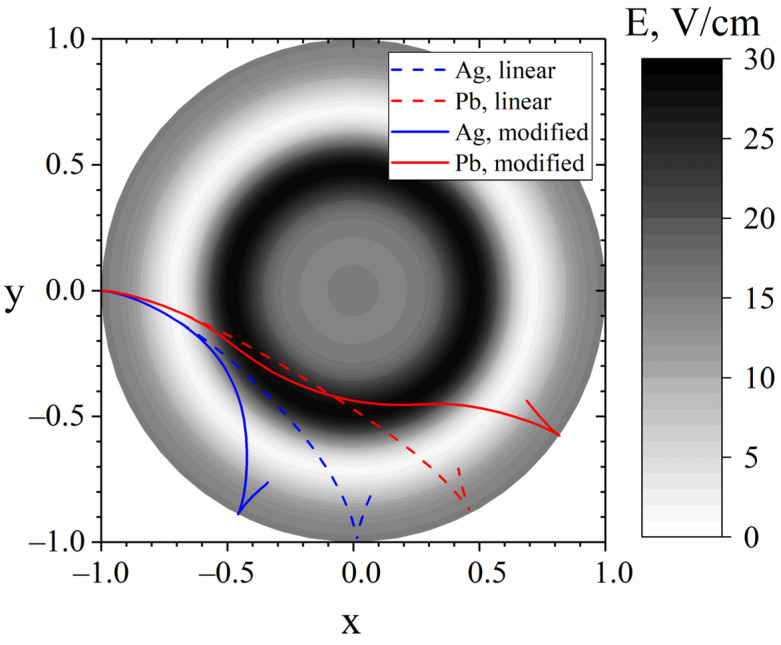
Trajectories of lead and silver in a linear potential with a field of 15 V/cm (dotted curves) and in a modified potential of type (15) at a=1, σ=0.12 and *ρ*_1_ = 0.525 (solid curves).

## Data Availability

Not applicable.

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
