# Peer review of "The Optimal Axis-Symmetrical Plasma Potential Distribution for Plasma Mass Separation"

_molecules, 2022, doi:10.3390/molecules27206824_

Round 1

Author Response

We thank the Reviewer for his careful reading of the article, providing missing actual list of citation and for his helpful comments. These comments helped to significantly improve the manuscript, and make the presentation of the material and conclusion clearer to a wide range of readers.

Below we provide answers to the comments.

Although the manuscript references a fair number of past studies on separation, I believe that the authors should expand the discussion and references to the literature of studies discussing the effect of a particular potential radial profile on the ion dynamics in cross-field geometry. This is indeed the core of their work and it would in my opinion be helpful to readers discovering this field. For instance:

  1. Ohkawa, T. and Miller, R. L., Phys. Plasmas (2002), 9, 5116
  2. Morozov, A. I. and Savel'ev, V. V. Plasma Phys. Rep. (2005), 31, 417
  3. Smirnov, V. P. and Samokhin, A. A. and Vorona, N. A. and Gavrikov, A. V., Plasma Phys. Rep. (2013), 39, 456
  4. Gueroult, R. and Rax, J.-M. and Fisch, N. J., Phys. Plasmas (2014), 21, 20701
  5. Bardakov, V. M. and Ivanov, S. D. and Strokin, N. A., Phys. Plasmas (2014), 21, 33505
  6. Gueroult R. et al., Plasma Phys. Control. Fusion (2018), 60, 14018

We agreed, that plasma separation of spent nuclear fuel topic and discussion of effect of a particular potential radial profile on the ion dynamics in cross-field geometry have not been fully developed. To expand the discussion of plasma mass separation with cross-field geometry we add the new paragraph (pages 2-3, lines 96-105) that presents small overview on current researches devoted to plasma separation topic including all provided references.

It is unclear to me what is meant here. More generally I think it would be

helpful it the authors could clarify what they mean by “Brillouin potential”.

Indeed, we have found, that the term “Brillouin potential” is not widespread. We clarify this term for the readers on page 6, lines 212-218. In Ref. 40 Brillouin introduced this potential as the electric potential of stationary rotating electron beam without radial electron movement. He considers the case, when electrons rotate in the beam’s bulk, and do not rotate on the periphery. So, his boundary conditions are similar to the Eqs. (3) and consequently, our problem considered is analogical, but we deal with positive charged particles.

The discussion seems to focus on the encouraging findings of Sec. 5 and not to mention the challenges identified in Sec. 4. Other than if the authors believe that small potential variations will no longer have a significant effect if on top a Gaussian modified profile (as per my question above), I would suggest to mention this explicitly in the conclusion. Although it poses challenges, I believe this is an important point to make.

We agreed, that results, described in Sec.4 and in the beginning of Sec. 5 are not discussed enough, and it is important to cover it in the Discussion. We found that challenges, identified in Sec 4. may have two sources. The first source is incorrect setting of the stationary potential profile parameters. Since we propose to establish a fairly wide potential plateau (zero electric field) in the region of the pericentre of the light component of the mixture, therefore, the expected inaccuracies associated with the positioning and potential value of this plateau will not lead to significant changes in the trajectories. This can be indirectly confirmed by Fig. 8 shows, that small variations of Gaussian parameter do not lead to significant effect on separation angle when using quite wide potential plateau (up to  the graph contains high gradient area). Another source of challenges connected with plasma potential fluctuations, which apparently disturb while real experiments. Unfortunately, fluctuations effect on separations depends on many parameters including shape, amplitude and period of deviation near pericenter, that makes analysis complicated and not available for quick assess. We expand the Discussion section analogical to the comments above.

Reviewer 2 Report

The present paper discusses analytical methods of devising an optimal control of mass separation using non-uniform electromagnetic fields in plasma. As far as I can tell, the paper is free of the major errors and the presented results are correct. The paper deals with, in my opinion, interesting and important topic in the modern-day plasma theory and applications.

However, the paper has several shortcomings. The most important one is that the list of citations, and how the cited papers are referenced, are strange. For example, all the cited papers about the Dusty Plasma, [2-7], are almost from the same one group, while there literally are dozens upon dozens of papers about the Dusty Plasma.
Similarly, papers by Gueroult and co-workers are mentioned by once - and only in relation to SNF. And results from other groups, such as Shinohara and co-workers, and Kaganovich and co-workers are not mentioned at all.

So for the paper to be published in Molecules, the Authors should significantly improve the list of citations, and clearly indicate how their results are relate to the current state of the art.

Author Response

We thank the Reviewer for his careful reading of the article and for his helpful comments. These comments helped to significantly improve the manuscript, and make the presentation of the material clearer to a wide range of readers.

Below we provide answers to the comments.

The most important one is that the list of citations, and how the cited papers are referenced, are strange. For example, all the cited papers about the Dusty Plasma, [2-7], are almost from the same one group, while there literally are dozens upon dozens of papers about the Dusty Plasma.

Indeed, we overuse Dusty Plasma citations in purpose to introduce the reader to topic of Colloidal systems. We deleted extra citations and diversified the paper by the citations on the works of the other groups.

Similarly, papers by Gueroult and co-workers are mentioned by once - and only in relation to SNF. And results from other groups, such as Shinohara and co-workers, and Kaganovich and co-workers are not mentioned at all.

We agreed, that plasma separation of spent nuclear fuel topic has not been fully developed. To expand the discussion of plasma mass separation with cross-field geometry we add the new paragraph (pages 2-3, lines 96-105) that presents small overview on current researches. We cited not only the scientific groups mentioned by the Reviewer, but also on a number of others.

Round 2

Reviewer 2 Report

The Authors adequately addressed my concerns. I my opinion, the manuscript has two shortcomings left: it is way too long (there are a lot of explanations of quite trivial algebraic steps) and the quality of English language should be improved.

Author Response

We thank Reviewer for his quick answer and his suggestions to improve the manuscript.

I my opinion, the manuscript has two shortcomings left: it is way too long (there are a lot of explanations of quite trivial algebraic steps) …

We agreed with Reviewer that manuscript contains several explanations with quite simple algebraic steps. Nevertheless, we think, that such trivial steps make the manuscript clear for wide range of readers. It may help young specialists to understand some mathematical details connected with current field. In purpose to shrink the manuscript we have already added Appendix A and Appendix B, which contains all extra information.

… and the quality of English language should be improved.

We revised the manuscript and improve the quality of English.